# Investigating Genetic Overlap between Alzheimer’s Disease, Lipids, and Coronary Artery Disease: A Large-Scale Genome-Wide Cross Trait Analysis

**DOI:** 10.3390/ijms25168814

**Published:** 2024-08-13

**Authors:** Artika Kirby, Tenielle Porter, Emmanuel O. Adewuyi, Simon M. Laws

**Affiliations:** 1Centre for Precision Health, Edith Cowan University, Joondalup, WA 6027, Australia; a.kirby@ecu.edu.au (A.K.); t.porter@ecu.edu.au (T.P.); 2Collaborative Genomics and Translation Group, School of Medical and Health Sciences, Edith Cowan University, Joondalup, WA 6027, Australia; 3Curtin Medical School, Curtin University, Bentley, WA 6102, Australia

**Keywords:** Alzheimer’s disease, coronary artery disease, gene-based analysis, global genetic correlation, lipids, linkage disequilibrium score regression, local analysis of [co]variant associations, local genetic correlation, Mendelian randomisation

## Abstract

There is evidence to support a link between abnormal lipid metabolism and Alzheimer’s disease (AD) risk. Similarly, observational studies suggest a comorbid relationship between AD and coronary artery disease (CAD). However, the intricate biological mechanisms of AD are poorly understood, and its relationship with lipids and CAD traits remains unresolved. Conflicting evidence further underscores the ongoing investigation into this research area. Here, we systematically assess the cross-trait genetic overlap of AD with 13 representative lipids (from eight classes) and seven CAD traits, leveraging robust analytical methods, well-powered large-scale genetic data, and rigorous replication testing. Our main analysis demonstrates a significant positive global genetic correlation of AD with triglycerides and all seven CAD traits assessed—angina pectoris, cardiac dysrhythmias, coronary arteriosclerosis, ischemic heart disease, myocardial infarction, non-specific chest pain, and coronary artery disease. Gene-level analyses largely reinforce these findings and highlight the genetic overlap between AD and three additional lipids: high-density lipoproteins (HDLs), low-density lipoproteins (LDLs), and total cholesterol. Moreover, we identify genome-wide significant genes (Fisher’s combined *p* value [*FCP_gene_*] < 2.60 × 10^−6^) shared across AD, several lipids, and CAD traits, including *WDR12*, *BAG6*, *HLA-DRA*, *PHB*, *ZNF652*, *APOE*, *APOC4*, *PVRL2*, and *TOMM40*. Mendelian randomisation analysis found no evidence of a significant causal relationship between AD, lipids, and CAD traits. However, local genetic correlation analysis identifies several local pleiotropic hotspots contributing to the relationship of AD with lipids and CAD traits across chromosomes 6, 8, 17, and 19. Completing a three-way analysis, we confirm a strong genetic correlation between lipids and CAD traits—HDL and sphingomyelin demonstrate negative correlations, while LDL, triglycerides, and total cholesterol show positive correlations. These findings support genetic overlap between AD, specific lipids, and CAD traits, implicating shared but non-causal genetic susceptibility. The identified shared genes and pleiotropic hotspots are valuable targets for further investigation into AD and, potentially, its comorbidity with CAD traits.

## 1. Introduction

Alzheimer’s disease (AD) is a prevalent neurodegenerative disorder characterised by cognitive decline and memory impairment [1,2]. The disorder represents a considerable public health challenge, with an anticipated global prevalence exceeding 139 million individuals by 2050 [3]. In Australia, dementia, predominantly AD, is the primary cause of disease burden among individuals aged 65 and older [4]. AD’s neuropathological features encompass hyperphosphorylated tau protein in neurofibrillary tangles and beta (β)-amyloid (Aβ) protein aggregation in senile plaques within the brain’s extracellular matrix [5,6]. Extensive research has focused on investigating AD’s aetiology and underlying biology [2,5,6,7,8,9], implicating various factors, including genetics, lifestyle, and environment [2,9,10]. Nevertheless, AD remains a multifaceted condition lacking curative treatments, thus posing a substantial global, social, and economic burden [10,11].

Lipid disorders and coronary artery disease (CAD) considerably impact human health [12,13,14]. Lipid disorders are recognised as a substantial risk factor for AD, just as a relationship between CAD and AD has been reported [15,16,17]. Lipids are vital in maintaining cell membrane structure, serving as energy reserves, and acting as signalling molecules [18]. In the brain, lipids are integral components of neuronal membranes and contribute substantially to various aspects of neuronal function, such as synaptic transmission, membrane dynamics, and intracellular signalling pathways [15,16,18]. However, abnormal lipid metabolisms have been implicated in AD pathogenesis, affecting factors such as blood–brain barrier integrity, amyloid precursor protein processing, myelination, receptor signalling, inflammation, oxidative stress, and energy imbalance [15,16,19]. Lipids also interact with genes, influencing gene-specific lipid proteins, enzymes, and metabolic pathways [15].

Recent studies, including those employing genome-wide association and cross-trait statistical analyses, have reported links or potential involvement of lipids in AD [15,16,20,21]. For instance, one study used the Mendelian randomisation (MR) method to reveal causal associations between specific lipid metabolites and AD [15]. Another study similarly explored polygenic associations between late-onset AD (LOAD) and blood lipid levels, demonstrating genetic concordance between AD and certain lipid metabolites [16]. These findings underscore the potential role of lipids in AD development but also suggest avenues for further investigation. CAD, characterised by the narrowing of the coronary arteries and plaque formation, has strong genetic components similar to AD [22]. CAD’s impact on the brain, including altered cerebrovascular function and blood–brain barrier disruption, provides conducive avenues for Aβ aggregation and potentially contributes to AD pathology [23]. 

Notably, observational evidence increasingly links CAD with cognitive impairment and the risk of dementia [24,25]. Research suggests that individuals with CAD experience an accelerated cognitive decline following diagnosis, with meta-analytical findings revealing a 26% higher relative risk of dementia among CAD patients [26,27]. Moreover, the onset of CAD at a younger age may exacerbate cognitive deterioration due to prolonged exposure to vascular lesions [26]. Despite these findings, the nature of the relationship and the underpinning mechanisms for CAD’s association with AD and cognitive impairment remains unclear. Given these premises and evidence for potential genetic overlap between CAD and AD, genetic analyses offer opportunities to further explore the shared mechanisms underlying these disorders [9,17]. 

Consistent with this position, a previous study has investigated the genetic determinants of CAD and their impact on the risk of LOAD [17]. The study utilised genome-wide association data, employing the MR analysis method (for causality testing), linkage equilibrium score regression (LDSC, for genetic correlation assessment), and GWAS-PW (a tool for jointly analysing two GWASs and scanning for shared genetic determinants) to assess the relationship between CAD and LOAD risk. The findings indicated a slightly higher risk of LOAD associated with a genetically determined risk of CAD [17]. However, when excluding the Apolipoprotein E (*APOE*) locus, the causal effect of CAD on LOAD risk was not significant [17]. The results highlight the predominant role of the *APOE* locus in the shared genetic architecture between CAD and LOAD, suggesting limited causal relevance of CAD to LOAD risk once *APOE* is considered [17]. The study underscores the need for further research to elucidate the potential shared genetic aetiology of CAD and LOAD. 

The connection between CAD and AD may partly reflect shared risk factors such as dyslipidaemia and inflammation. However, there is also the potential for shared genetic predispositions. For example, the *APOE* gene is well-established in AD, with individuals carrying the ε4 allele at a higher risk of AD [28,29,30,31]. These individuals may also be more likely to develop CAD due to their lipid profile and other related factors. Although *APOE* is more prominently associated with AD, its role in lipid metabolism makes it relevant to understanding CAD [17], underscoring the interconnectedness of genetic influences on different aspects of cardiovascular and neurological health. Indeed, various lipids are associated with cardiovascular diseases (CVD), and lipids likely influence the risk of dementia through their association with CVD [32,33]. Furthermore, AD and CAD (or CVD, more broadly) have been shown to share a range of pleiotropic genes, including *GPBP1*, *SETDB2*, *DAB2IP*, and *DNM2* [34,35]. Genetic variants such as rs116426890-T and rs62118504-G have also been implicated [35]. The variant rs116426890-T is linked to the expression of genes, including *ABI2*, *CARF*, *ICA1L*, *FAM117B*, and *NBEAL1*, which are associated with various cardiovascular traits and potentially AD pathology [35]. Meanwhile, rs62118504-G is mapped to *EXOC3L2* and *MARK4*, which are linked to AD [35]. 

Thus, existing evidence supports a potential link between AD, lipids, and CAD putatively through shared genetic susceptibility [15,16,17]. However, the relationships and underlying biological mechanisms remain unresolved despite several years of enquiry [15,16,17,33,36]. Additionally, contrasting reports exist [33,37,38], just as the evidence regarding the genetic overlap of AD with lipids and CAD traits is inconclusive [15,16,17,36]. Although AD currently has no curative treatments, disentangling the effects of abnormal lipid metabolism and CAD traits on its risk (and vice versa) will have substantial implications for advancing knowledge of its underlying biology and identifying potential therapeutic targets for further investigation. 

Our study employs a three-way cross-traits genetic analysis approach, focusing on the interplay between lipids (comprising 13 representative traits from eight lipid classes), CAD, and various CAD-related traits in relation to AD. Building upon previous research, we investigate these relationships by leveraging increasingly powerful analytical methods and well-powered datasets. Moreover, we examine the genetic overlap of AD with lipids and CAD traits at single-nucleotide polymorphism (SNP) and gene-based levels to provide a robust insight since genes are closer to biology than SNPs. Importantly, we identify pleiotropic loci (using the local genetic correlation approach) and shared genes (using gene-based association analysis) across AD, lipids, and CAD traits. Our research offers further insights through a more comprehensive but targeted focus on AD (with rigorous [partial] replication testing), specific representative lipids, and an array of CAD traits (i.e., not just CAD but several related phenotypes). This study provides robust evidence, advances our understanding of the intricate genetic connections between AD, lipids, and CAD-related traits, and identifies potential targets for further investigation. 

## 2. Results

Figure 1 presents a simplified workflow for this study. First, using the LDSC method [39], we assessed and quantified SNP-level pairwise global (genome-wide) genetic correlations between 13 lipids, seven CAD traits, and AD. Second, we conducted bi-directional two-sample Mendelian randomisation (2SMR) [40] analyses to test for potential causal associations between lipids, CAD traits, and AD. Third, we performed gene-based analyses [41] and subsequently assessed gene-level genetic overlap of lipids and CAD traits with AD. Using the results of our gene-based association analyses, we identify genome-wide significant (GWS, sentinel) genes shared by AD, lipids, and CAD traits. Also, following the practice employed in previous studies [19,42,43,44,45,46,47,48,49], we applied Fisher’s combined *p*-value (FCP) method to identify shared genes reaching GWS for AD, lipids, and CAD traits. Lastly, we assessed the local genetic correlations between lipids, CAD traits, and AD [50]. 

### 2.1. Global Genetic Correlation of AD with Lipids and CAD Traits

Table 1 provides an overview of the GWAS datasets employed for the analysis of the relationship between AD, lipids, and CAD traits. Further specific information for each cohort can be found in Appendix A. Table 2 represents the results of the genome-wide genetic correlation estimates between AD, lipids, and CAD traits using the LDSC analysis method.

In our first round of analyses, we assessed the global genetic correlation between AD (Jansen et al. [28]) and 13 lipids, with a pairwise testing correction, considered significant at *p* ≤ 0.025. Only one of the lipids (triglycerides) reached a significant status in its genetic relationship with AD, demonstrating a positive global genetic correlation (rg = 0.09, se = 0.04, *p* = 1.09 × 10^−2^). The results for the association of other lipids with AD reveal no significant global genetic correlation, as summarised in Table 2. For a possible replication of our findings, we further assessed the global genetic correlation between each of the 13 lipid traits and the AD GWAS from Lambert et al. [29]. This analysis reveals significant positive global genetic correlations between AD and three lipid traits, including low-density lipoproteins (LDLs) (rg = 0.12, se = 0.05, *p* = 1.87 × 10^−2^), triglycerides (rg = 0.08, se = 0.04, *p* = 2.46 × 10^−2^), and total cholesterol (rg = 0.12, se = 0.05, *p* = 1.55 × 10^−2^) (Appendix A).

Secondly, we assessed the global genetic correlation between the AD GWAS (from Jansen et al. [28]) and seven CAD traits, all of which demonstrated a significant correlation that survived the pairwise testing correction (ρ≤ 0.025) in our LDSC analysis. The global genetic correlations were significant and positive between AD and angina pectoris (rg = 0.21, se = 0.04, *p* = 5.88 × 10^−9^), cardiac dysrhythmias (rg = 0.14, se = 0.04, *p* = 3.49 × 10^−4^), coronary arteriosclerosis (rg = 0.17, se = 0.03, *p* = 2.26 × 10^−8^), ischemic heart disease (rg = 0.20, se = 0.03, *p* = 1.39 × 10^−10^), myocardial infarction (rg = 0.17, se = 0.04, *p* = 1.03 × 10^−5^), non-specific chest pain (rg = 0.22, se = 0.04, *p* = 2.06 × 10^−8^), and CAD (rg = 0.15, se = 0.04, *p* = 3.74 × 10^−4^). Using the AD GWAS from Lambert et al. [29] as a partial replication set, we found a significant genetic correlation with two of the CAD traits at the nominal level of significance, including CAD (rg = 0.10, se = 0.05, *p* = 2.78 × 10^−2^) and non-specific chest pain (rg = 0.07, se = 0.04, *p* = 4.15 × 10^−2^) (Appendix A). Additional details of the genetic correlation analyses are available in the Appendix A.

Thirdly, completing a three-way assessment, we investigated the global genetic correlation between each CAD trait and the 13 lipid traits (Table 3). Overall, significant global genetic correlations were observed between CAD traits and lipids—high-density Lipoproteins (HDLs), LDL, triglycerides, total cholesterol, and 154 SM C16:1 sphingomyelin (Table 3 and Appendix A). Specifically, we found a significant and negative global genetic correlation between HDL and all seven CAD traits, with the most significant result evidenced between CAD and HDL (rg = −0.37, se = 0.04, *p* = 6.52 × 10^−17^) (Table 3). Interestingly, a significant negative global genetic correlation was also evidenced between coronary arteriosclerosis and 154 SM C16:1 sphingomyelin (rg = −0.31, se = 0.12, *p* = 1.18 × 10^−2^) (Table 3). 

Furthermore, we found a significant positive global genetic correlation between each of the seven CAD traits and triglycerides, including angina pectoris (rg = 0.41, se = 0.06, *p* = 8.92 × 10^−13^), cardiac dysrhythmias (rg = 0.14, se = 0.04, *p* = 3.70 × 10^−4^), coronary arteriosclerosis (rg = 0.37, se = 0.05, *p* = 2.11 × 10^−15^), ischemic heart disease (rg = 0.40, se = 0.05, *p* = 2.99 × 10^−15^), myocardial infarction (rg = 0.41, se = 0.06, *p* = 2.82 × 10^−13^), non-specific chest pain (rg = 0.31, se = 0.05, *p* = 7.10 × 10^−9^), and CAD (rg = 0.42, se = 0.04, *p* = 1.41 × 10^−21^) (Table 3). We observed similar significant and positive global genetic correlations between LDL and all the CAD traits except cardiac dysrhythmias (rg = 0.02, se = 0.03, *p* = 4.66 × 10^−1^) (Appendix A). Additional details of the global genetic correlation analyses between lipids and CAD traits are presented in Appendix A.

### 2.2. Results of Gene-Level Genetic Overlap Analysis

We performed gene-based analysis to further assess the genetic overlap, at the gene level, of lipids and CAD traits with AD. We restricted our gene-based analysis to include lipids that were significant in our initial LDSC analyses (either main or validation set), including HDL, LDL, triglycerides, and total cholesterol, and all the seven CAD traits–angina pectoris, cardiac dysrhythmias, coronary arteriosclerosis, ischemic heart disease, myocardial infarction, non-specific chest pain, and CAD. We tested the relationship of these traits with AD to investigate whether they shared genetic components more than by chance. 

In our main analysis, we obtained the total number of SNPs overlapping between AD and each lipid and CAD trait to ensure that we performed an equivalent gene-based analysis (Appendix A). A total of 2,402,800 SNPs overlapped between AD and HDL, 2,394,019 with LDL, 2,395,509 with triglycerides, 2,402,374 with total cholesterol, and 2,393,041 with CAD. Similarly, 11,250,658 AD SNPs overlapped with angina pectoris, 11,251,568 with cardiac dysrhythmias, 11,250,900 with coronary arteriosclerosis, 11,251,741 with ischemic heart disease, 11,250,320 with myocardial infarction, and 11,251,773 with non-specific chest pain. Following the gene-based association analysis, 18,960 protein-coding genes were identified for angina pectoris, cardiac dysrhythmias, coronary arteriosclerosis, ischemic heart disease, myocardial infarction, and non-specific chest pain (Table 4). We also identified 17,735 protein-coding genes for non-specific chest pain, 17,683 for each of HDL and total cholesterol, 17,671 for triglycerides, and 17,669 for LDL.

To perform gene-level overlap assessment, we assigned lipids and CAD traits as the discovery sets and AD as the target (Table 4). We identified the total number of genes associated with each trait in the discovery and target sets at P_gene_ < 0.05 (Table 4). Accordingly, at P_gene_ < 0.05, 1880 genes were associated with HDL, 1766 with LDL, 1743 with triglycerides, 1988 with total cholesterol, 2175 with angina pectoris, 1776 with cardiac dysrhythmias, 2524 with coronary arteriosclerosis, 2710 with ischemic heart disease, 1995 with myocardial infarction, 1943 with non-specific chest pain, and 1601 with CAD. We reported the number of genes associated with AD at each corresponding analysis pair (Table 4). Moreover, at P_gene_ < 0.05, we identified genes overlapping the target and corresponding discovery sets with a total of 294 AD genes overlapping with HDL, 267 with LDL, 273 with triglycerides, 320 with total cholesterol, 260 with angina pectoris, 212 with cardiac dysrhythmias, 333 with coronary arteriosclerosis, 315 with ischemic heart disease, 244 with myocardial infarction, 212 with non-specific chest pain, and 201 with CAD. To assess gene-level genetic overlap, we compared the expected proportion of gene overlap, at P_gene_ < 0.05, with the observed proportion of gene overlap (see methods for additional details). 

The results of the exact binomial test support a significant gene-level genetic overlap, at P_gene_ < 0.05, between AD and all four lipids including HDL (P_binomial-test_ [P_b-t_] = 9.84 × 10^−15^), LDL (P_b-t_ = 1.28 × 10^−11^), triglycerides (P_b-t_ = 2.24 × 10^−13^), and total cholesterol (P_b-t_ = 2.20 × 10^−16^). Similarly, we found a significant gene-level overlap between AD and CAD traits including angina pectoris (P_b-t_ = 3.65 × 10^−4^), cardiac dysrhythmias (P_b-t_ = 1.48 × 10^−3^), coronary arteriosclerosis (P_b-t_ = 4.73 × 10^−9^), ischemic heart disease (P_b-t_ = 4.60 × 10^−4^), myocardial infarction (P_b-t_ = 1.18 × 10^−4^), and CAD (P_b-t_ = 6.26 × 10^−4^). Non-specific chest pain did not survive pairwise testing correction (*p* < 0.025) in gene-level overlap with AD. However, it was significant at the nominal level (P_b-t_ = 3.69 × 10^−2^) (Table 4). 

### 2.3. Genome-Wide Significant (Sentinel) Genes Shared by AD, Lipids, and CAD Traits

Our gene association analysis identified GWS genes (P_gene_ < 2.60 × 10^−6^, that is, genes that were already GWS in our dataset, ‘sentinel genes’) across AD, lipids, and each of the CAD traits (Appendix A). AD had 66 GWS genes (P_gene-AD_ < 2.60 × 10^−6^, Appendix A). For CAD traits, coronary arteriosclerosis had the highest number of GWS genes at 68 (P_gene-coronary-arteriosclerosis_ < 2.60 × 10^−6^, Appendix A), then ischemic heart disease with 53 (P_gene-ischemic-heart-disease_ < 2.60 × 10^−6^, Appendix A), followed by cardiac dysrhythmias with 42 (P_gene-cardiac-dysrhythmias_ < 2.60 × 10^−6^, Appendix A), and myocardial infarction with 38 (P_gene-myocardial-infarction_ < 2.60 × 10^−6^, Appendix A), then angina pectoris with 24 (P_gene-angina-pectoris_ < 2.60 × 10^−6^, Appendix A), and CAD with 11 (P_gene-CAD_ < 2.60 × 10^−6^, Appendix A), and, lastly, non-specific chest pain with 9 (P_gene-non-specific-chest-pain_ < 2.60 × 10^−6^, Appendix A). There were more GWS genes for lipids, with the highest number at 216 for total cholesterol (P_gene-Total-cholesterol_ < 2.60 × 10^−6^, Appendix A), followed by HDL with 172 genes (P_gene-HDL_ < 2.60 × 10^−6^, Appendix A), and 154 GWS genes for both LDL (P_gene-LDL_ < 2.60 × 10^−6^, Appendix A) and triglycerides (P_gene-Triglycerides_ < 2.60 × 10^−6^, Appendix A). 

Assessing overlap between GWS (P_gene_ < 2.60 × 10^−6^) genes for AD, lipids, and CAD traits, that is, sentinel genes; we found two (*APOE* and *ZNF652*) that were shared by AD and angina pectoris (Appendix A), five (*APOC1*, *APOE*, *PVRL2*, *TOMM40*, and *ZNF652*) by AD and both coronary arteriosclerosis (Appendix A) and ischemic heart disease (Appendix A), three (*APOC1*, *APOE*, and *TOMM40*) by AD and myocardial infarction (Appendix A), and five (*APOC4*, *APOC4-APOC2*, *APOE*, *CTB-129P6.11*, and *TOMM40*) by AD and HDL (Appendix A). Additionally, 9 GWS genes overlap between AD and triglycerides (Appendix A), followed by an overlap of 11 GWS genes between AD and total cholesterol (Appendix A) and 16 between AD and LDL (Appendix A). We did not observe a GWS gene (sentinel) overlap between AD and cardiac dysrhythmias, non-specific chest pain, and CAD. Table 5 (the upper section) summarises GWS genes (sentinel) overlapping AD and two or more lipids or CAD traits.

### 2.4. Shared Genes Reaching Genome-Wide Significance for AD, Lipids, and CAD Traits

Given their significant SNP-level global genetic correlation and gene-based genetic overlap, we performed a further assessment using the FCP method [19,42,43,44,45,46,47,48,49] to identify genes shared by AD, lipids, and CAD traits. We identified a range of gene overlap between AD and each lipid and CAD trait, many of which reached GWS (P_gene-AD_ < 2.60 × 10^−6^) or with evidence of improvement following the FCP analysis, as seen in Appendix A. We followed up on this analysis, aiming to identify genes that were not previously GWS (based on our data) in AD (0.05 < P_gene-AD_ > 2.60 × 10^−6^) or lipids (0.05 < P_genes-lipids_ > 2.60 × 10^−6^) or CAD traits (0.05 < P_gene-CAD-traits_ > 2.60 × 10^−6^) but reached the status following FCP analysis (P_FCP_ < 2.60 × 10^−6^). For this analysis, we identified nine genes (CARF, CKM, HLA-DQB1, HLA-DRA, HLA-DRB1, ICA1L, PLCG2, TMEM106B, and WDR12) reaching GWS for AD and angina pectoris (Appendix A). Two genes (DOC2A and PHB) reached GWS for AD and cardiac dysrhythmias (Appendix A). Coronary arteriosclerosis shared 10 GWS genes with AD (BAG6, C6orf10, HLA-DQB1, HLA-DRA, HLA-DRB1, NDUFAF6, NME7, PLCG2, PRRC2A, and TRIB1), while myocardial infarction shared three genes with AD (HLA-DRA, PHB, and RP11-81K2.1) [Appendix A]. Other genes reaching GWS across AD and each lipid or CAD trait are presented in Appendix A. Table 5 (the lower section) summarises genes reaching GWS in the FCP analysis across AD and at least two lipids or CAD traits. 

### 2.5. Results of Causal Relationship Assessment

We performed bi-directional 2SMR analyses to test for a potential causal association of selected lipids and all seven CAD traits with AD. We restricted our 2SMR analysis to only four lipids, including HDL, LDL, triglycerides, and total cholesterol. We included LDL, triglycerides, and total cholesterol as they were observed to have a significant global genetic correlation with AD in the main or validating analysis. Additionally, due to the relationship between HDL and AD underscored in a recent publication [31], we included HDL to further explore its potential causal associations with AD. We included all seven CAD traits listed in this study based on their strong global genetic correlations with AD. 

#### 2.5.1. No Causal Relationship of Lipids with Alzheimer’s Disease

Our 2SMR assessment found no evidence of a significant causal association between lipids—HDL, LDL, triglycerides, and total cholesterol—as the exposure and AD as the outcome variable (Table 6). In reverse analyses, in which AD was assessed as the exposure variable against each lipid as an outcome variable, our findings indicate that genetic liability to AD had no significant causal effect on any of the lipids (Table 6). The results were consistent across other MR models, including the weighted-median and MR-Egger models (Table 6). We also did not observe any evidence of significant pleiotropy or heterogeneity in our 2SMR analysis. When using the MR-PRESSO method, the raw output replicated similar IVW-based non-significant results for lipids—LDL, triglycerides, and total cholesterol—except for HDL, which was not observed with an MR-PRESSO assessment, potentially a reflection of too few instrumental variables (nIV = 3) (Table 6). There was no evidence of outlier-corrected results based on removing potentially pleiotropic SNPs for lipids—LDL, triglycerides, and total cholesterol. Additional details of the causal relationship of lipids with AD are presented in Appendix A.

#### 2.5.2. No Causal Relationship of CAD Traits with Alzheimer’s Disease

We first tested the causal association between CAD traits and AD, with CAD traits as the exposure variables. We found no evidence of a significant causal effect of the seven CAD traits on AD risk. We note an absence of results for non-specific chest pain as the exposure variable on AD risk, with only one instrumental variable present for the 2SMR analysis. These results were consistent across the IVW, weighted-median, MR-Egger, and MR-PRESSO analyses (Table 6). Similarly, by changing the direction of our analysis, we found no evidence of a significant causal effect between AD as the exposure and each of the seven CAD traits assessed as outcome variables in our study. These results were replicated across other MR models, including the weighted-median and MR-Egger models. We also observed that MR-PRESSO did not produce output corrected for potential pleiotropy for the bi-directional relationship of all seven CAD traits and AD, indicating no outlier instruments to correct. Additional details of the causal relationship between CAD traits and AD are presented in Appendix A.

### 2.6. Local Genetic Correlation of Alzheimer’s Disease with Lipids and CAD Traits

We used LAVA [50] to perform local genetic correlation analyses of lipids and CAD traits with AD. This approach enabled us to identify disproportionate genetic correlations in specific genomic regions between lipids, CAD traits, and AD. Unlike the LDSC method, which takes an average of the correlation across the whole genome (global or genome-wide correlation) [39], the LAVA approach provides insights into the local effects and shared genetic basis between two or more traits [50]. At the threshold of ρ≤ 1.40 × 10^−3^, adjusting for multiple testing, LAVA detected 18 significant bivariate local genetic correlations across eight loci, contributing to the relationship of AD with lipids and CAD traits (Table 7 and Figure 2). The identified loci spread across chromosomes 6, 8, 17, and 19, including 962 (chr6: 32,208,902–32,454,577) [AD–LDL], 963 (chr6: 32,454,578–32,539,567) [AD–cardiac dysrhythmias], 964 (chr6: 32,539,568–32,586,784) [AD–LDL, AD–total cholesterol, and AD–myocardial infarction], 965 (chr6: 32,586,785–32,629,239) [AD–angina pectoris], 966 (chr6: 32,629,240–32,682,213) [AD–LDL and AD–total cholesterol], 1351 (chr8: 125,453,323–126,766,827) [AD–total cholesterol], 2209 (chr17: 45,883,902–47,516,224) [AD–ischemic heart disease], and 2351 (chr19: 45,040,933–45,893,307) [AD–HDL, AD–LDL, AD–total cholesterol, AD–triglycerides, AD–angina pectoris, AD–coronary arteriosclerosis, AD–ischemic heart disease, and AD–myocardial infarction] (Table 7 and Figure 2). Notably, the locus at 2351 (chr19: 45,040,933–45,893,307) was the most implicated in several traits, and the direction was positive in each of the analyses except in AD–HDL, where the effect direction was negative (Table 7).

We further assessed the chromosome and loci for each significant pairwise correlation hotspot (Table 7 and Figure 2). Chromosome 19 and Chromosome 6 were each observed with the highest number (n = 8) of pairwise trait associations (Figure 2). On chromosome 19, pairwise trait associations were observed at a singular locus region (chr19: 2351 [45,040,933–45,893,307]), whereas on chromosome 6, associations were observed across multiple locus positions (chr 6: 962 [32,208,902–32,454,577], 963 [32,454,578–32,539,567], 964 [32,539,568–32,586,784], 965 [32,586,785–32,629,239], and 966 [32,629,240–32,682,213]) (Figure 2).

Additionally, for lipids, including AD–LDL, we found local genetic correlations on chromosome 6 (chr6: 962 [32,208,902–32,454,577], 964 [32,539,568–32,586,784], and 966 [32,629,240–32,682,213]), and chromosome 19 (chr19: 2351 [45,040,933–45,893,307]) (Table 7 and Figure 2). For AD–total cholesterol, we observed local correlations on three chromosomes including chromosome 6 (chr 6: 964 [32,539,568–32,586,784], and 966 [32,629,240–32,682,213]), chromosome 8 (chr 8: 1351 [125,453,323–126,766,827]), and chromosome 19 (chr19: 2351 [45,040,933–45,893,307]) (Table 7 and Figure 2). On the other hand, for CAD traits, for AD–angina pectoris we observed local correlations on chromosome 6 (chr6: 965 [32,586,785–32,629,239]), for AD–cardiac dysrhythmias on chromosome 6 (chr6: 963 [32,454,578–32,539,567]), and for AD–ischemic heart disease on chromosome 17 (chr17: 2209 [45,883,902–47,516,224]). Both AD–angina pectoris and AD–ischemic heart disease were also observed on chromosome 19 (chr19: 2351 [45,040,933–45,893,307]) (Table 7 and Figure 2). Table 7 and Figure 2 summarise other identified loci.

### 2.7. Comparing LDSC and LAVA Results 

To compare the global and local genetic correlation results across the pairs of traits assessed, we utilised the ‘rg’ (genetic correlation) estimates for LDSC and the ‘Mean.RHO’ (average of RHO estimates) from LAVA. We then assessed whether these measures met their respective pairwise testing correction cutoffs to determine if they had significant differences. (Table 2 and Table 7 and Figure 3). Firstly, we examined whether any significant correlations identified in LDSC were absent in LAVA and vice versa. In our primary analysis, following adjustments for pairwise testing, we detected a positive correlation between AD and ‘non-specific chest pain’ using LDSC (rg = 0.22, *p* = 2.06 × 10^−8^, see Table 2). Conversely, in LAVA, the correlations between these traits were only marginally significant and did not survive correction for pairwise testing (*p* = 1.43 × 10^−3^, see Figure 3 and Appendix A). Likewise, employing LDSC, we identified a significant positive correlation between AD and CAD (rg = 0.15, *p* = 3.74 × 10^−4^, see Table 2); however, in LAVA, these correlations were only significant at the nominal level (see Figure 3 and Appendix A). On the other hand, with LAVA, following adjustments for pairwise testing correction, we identified a significant negative correlation between AD and HDL (Mean.RHO = −0.29, *p* = 3.75 × 10^−10^, see Table 3) and a significant and positive correlation between AD and LDL (Mean.RHO = 0.52, see Table 3) and AD and total cholesterol (Mean.RHO = 0.40, see Table 3); however, these pairwise traits were evidenced as non-significant in our LDSC analysis (see Appendix A).

Secondly, we investigated concordance in the direction of effect between LDSC and LAVA. For example, we assessed whether the observed positive correlation in LDSC is consistent with LAVA’s directionality, whether LAVA shows an opposing trend or did not detect any significant correlation. For AD and cardiac dysrhythmias, LDSC found a significant positive correlation (rg = 0.14, *p* = 3.49 × 10^−4^); however, LAVA analysis represented an opposing output, being a significant negative correlation (Mean.RHO = −0.38, *p* = 7.25 × 10^−6^) (Figure 3). 

## 3. Discussion

Research has long suggested that lipids play a role in AD [15,16,18,19]. Similarly, observational studies indicate a potential comorbid relationship between AD and CAD traits [22,23]. Evidence also shows that lipids are implicated in CAD traits [12,13,14]. However, the nature of AD’s association with lipids and CAD traits remains largely unresolved [9,17,54]. In this study, we performed a comprehensive analysis to systematically assess the genetic relationship of AD [28,29] with 13 lipids (from eight representative classes) [12,51,52] and seven CAD traits, using well-regarded and advanced statistical genetic analytic tools [39,40,41,42,43,44,45,46,47,48,50]. Our study reveals noteworthy findings, providing new insights into the complex relationships between these traits.

Our comprehensive LDSC analysis (discovery and validation sets) reveals a significant and positive global genetic correlation between AD and three lipid traits: LDL, triglycerides, and total cholesterol. These correlations suggest that, at the least, a proportion of individuals with a genetic predisposition to elevated levels of the lipids may have an increased risk of AD. However, not all lipid traits were significantly associated with AD. For example, we did not observe a significant global genetic correlation between AD and HDL, although our gene-level overlap analysis uncovered this association. This selective association suggests that certain lipid traits may have a more direct genetic link with AD, potentially through specific pathways. In contrast, many lipids assessed were genetically correlated with CAD traits. Specifically, we found significant genetic correlations between CAD traits and lipid profiles such as HDL, LDL, triglycerides, and total cholesterol, aligning with the existing literature that underscores the genetic interplay in cardiovascular risk factors [55,56]. The negative correlation between HDL and CAD traits is consistent, generally (‘generally’ because a U-shaped relationship is also known [57]), with studies highlighting HDL’s protective role against CVD, while the positive correlations observed between triglycerides and CAD traits reinforce the view that elevated triglyceride levels are a significant risk factor for cardiovascular events [56]. 

Interestingly, our study also reveals a negative correlation between coronary arteriosclerosis and 154 SM C16:1 sphingomyelin, suggesting potential unexplored pathways involving lipid metabolism that warrant further investigation. Contrasting findings have been reported on the role of various sphingolipids on CVD risk [58,59,60]. Our current findings, however, provide genetic evidence supporting the potential association of the 154 SM C16:1 sphingomyelin with a reduced risk of a CAD trait while highlighting the complexity of lipid interactions in cardiovascular pathology. The finding that all CAD traits were positively correlated with AD (supported strongly by our gene-level genetic overlap results), despite only some lipids being associated with AD, suggests a possible indirect pathway linking some lipids to AD through CAD traits. It is conceivable that the genetic factors contributing to CAD traits, which are closely associated with a broad range of lipid profiles, also increase the risk of AD. This observation could indicate a shared genetic architecture or common risk factors that predispose individuals to both CAD and AD.

Our global genetic correlation findings contrast with some studies, such as those of Zhu et al. [61], which evaluated the genetic correlation between AD and specific lipid traits using the LDSC approach. Unlike Zhu et al. [61], we found no significant correlation between AD and HDL (although our gene-level analysis made this discovery). Additionally, our study uncovered a notable positive correlation between AD and other lipids, including LDL, triglycerides, and total cholesterol, which were not identified in Zhu et al.’s analysis [61]. Furthermore, Grace et al. [17] reported no genetic correlation between the two conditions in their LDSC cross-trait analysis between AD and CAD. In contrast, our investigation revealed a significant positive genetic correlation of AD with CAD and all CAD-related traits evaluated in our study (not just CAD). These disparities may arise from variations in the datasets analysed, including differences in sample sizes. In the current study, we conducted a comprehensive assessment by integrating additional data, utilising relatively larger sample sizes, and assessing the potential partial replication of our results, thereby enhancing the robustness of our investigation. 

To deepen our understanding of the relationship between AD, lipids, and CAD traits, we conducted bi-directional 2SMR analyses. Our findings did not support a causal link between AD, lipids, and CAD traits, regardless of the direction of analysis—whether AD was used as the exposure or outcome variable. Our results remained consistent across various sensitivity tests and MR models, including the MR-Egger intercept for a pleiotropy test, the MR-Egger model, the weighted median, and the MR-PRESSO method. These outcomes partially align with the findings of Grace et al. [17], where CAD was causally associated with a 7% increased LOAD risk, primarily attributed to variants in the *APOE* region. However, the association dissipated upon excluding the *APOE* region [17]. In contrast, our analysis encompassed all IVs satisfying the MR assumptions in examining the relationship between AD and the seven CAD traits—not solely CAD as assessed by Grace et al. [17]. Our findings revealed no evidence of a causal association between AD and CAD traits. Thus, while AD is genetically correlated with CAD (as observed in our study but not in Grace et al. [17]), MR results suggest that causal inference cannot explain their relationship. Notably, a recent study [37] reported a significant causal association between AD and HDL. This finding was not evident in our study, and the reported direction of causation contradicts the negative genetic correlation observed between AD and HDL by others [17,61], revealing inconsistent findings, which may be explained by the datasets analysed and the U-shape relationship of HDL [57], suggesting that the lipid is not a homogeneous entity. The lipid consists of various subtypes with potentially different functions. The protective effects of HDL might depend on its functionality rather than its concentration alone.

Our analysis delves deeper into understanding the correlations at a local level, focusing on specific genomic loci [39,50]. Even in instances where no global genetic correlation was observed, correlations may still exist at specific genomic regions, thus underscoring the focus of our analysis. In this regard, we utilised a newly developed and potentially more powerful analysis tool, LAVA [50], and our investigation yielded essential insights. For instance, we identified 18 significant bivariate local genetic correlations across eight loci, contributing to the association between AD, lipids, and CAD traits. These loci were distributed across chromosomes 6, 8, 17, and 19. Notably, the locus at 2351 (chr19: 45,040,933–45,893,307) exhibited significant involvement across multiple traits, consistently showing a positive correlation in all analyses except for AD–HDL, which demonstrated a negative association. 

Comparing LAVA and LDSC results, we observed, firstly, concordance between the global and local genetic correlation for AD and lipid traits, namely HDL, LDL, triglycerides, and total cholesterol, as well as for CAD traits, including angina pectoris, coronary arteriosclerosis, ischemic heart disease, and myocardial infarction, providing additional evidence of shared genetic components of AD with these traits [39,50]. Our gene-level genetic overlap assessment further supported and highlighted these findings [41,42,43]. Of note, Zhu et al. [61] utilised p-Hess for their local genetic correlation analyses and reported only one significant region, on chromosome 19, for the local genetic correlation between AD and LDL. In our study, using LAVA for the local genetic correlation between AD and LDL, we found a significant hotspot on chromosome 19, as well as multiple loci (962, 964, and 966) on chromosome 6. LAVA has previously been shown (through extensive simulations) to demonstrate well-controlled type 1 error and superior performance over existing approaches, including p-Hess [50,62]. Hence, we consider our findings robust, providing new insights into the relationship of these traits. 

Secondly, our findings revealed a discordance between the results of LDSC and LAVA regarding the correlation between AD and cardiac dysrhythmias, particularly regarding the direction of the effects. While LDSC suggested a positive genetic correlation, LAVA revealed a negative association at several identified loci. Additionally, discordance was observed between AD and other CAD traits, such as non-specific chest pain and CAD. For example, while LDSC indicated a significant and positive global genetic correlation, LAVA’s results for these pair of traits did not survive pairwise testing correction (although nominally significant). These findings remained largely consistent in the replication set. The discrepancies observed between LDSC and LAVA results underscore the intricate nature of genetic interactions and emphasise the importance of employing multiple analytical approaches when investigating genome-wide genetic correlations [42,43,63]. Furthermore, our LAVA analysis revealed that several of the identified pleiotropic loci were at chromosome 19 and chromosome 6, indicating that these are the local genomic loci contributing more disproportionately to the relationship between AD, lipids, and CAD traits. On the other hand, we found local genetic correlations for AD—total cholesterol (chr8: locus 1351) and for AD—ischemic heart disease (chr17: locus 2209), indicating that these hotspots represent important targets for further investigation in AD and the respective CAD traits.

We performed further analysis to identify GWS genes (sentinel genes) overlapping between AD, lipids, and CAD traits in our datasets. Consistent with previous studies [14,35,64], our study revealed that *APOE*, *TOMM40*, and many genes on chromosome 19 are shared across AD, multiple lipids (such as HDL, LDL, triglycerides, and total cholesterol), and CAD traits (including coronary arteriosclerosis, myocardial infarction, and ischemic heart disease). Further, we assessed genes reaching GWS following our FCP analysis. Findings in this analysis revealed multiple genes not previously GWS for AD, lipids, or CAD traits but reached the status, potentially indicating them to be putatively novel genes (based on our data); there is a possibility they have been identified in other GWAS data, which would, in essence, validate our finding. Also, we identified a GWS sentinel gene on chromosome 17 and others reaching the status through the FCP analysis on chromosomes 2, 6, 7, 8, and 16. Our findings for these GWS genes agree largely with our LAVA local genetic correlation results, which revealed chromosomes 19 and 6 as major hotspots for AD and the named lipids and CAD traits.

Our study is notable for utilising a comprehensive suite of complementary and highly regarded statistical genetic analysis models. These methods were rigorously employed to assess the intricate relationship between AD, selected representative lipids, and CAD traits, offering new insights into the complex interplay among these traits. Nonetheless, our study has limitations that warrant careful consideration when interpreting its findings. Firstly, the GWAS data utilised are from individuals of European ancestry; caution is advised when extrapolating or comparing our findings to populations of diverse ancestries. Secondly, while sample overlap is known to confound some of the analyses (for example, genetic correlation and MR analyses), our preliminary assessment indicates no evidence for substantial overlap of samples between AD and the traits assessed. Lastly, despite our study’s inability to establish a significant causal effect between AD and the traits examined, we acknowledge the possibility of such a relationship. Notably, some GWAS datasets employed in our study may be limited by the number of instrumental variables available, potentially influencing our present results. Consequently, future research endeavours leveraging more robust GWAS datasets, not necessarily based on sample size [65] (as they become available), are worthwhile to further elucidate and refine the causal relationships observed in our study.

## 4. Materials and Methods

### 4.1. Data Sources

We assessed the relationship between AD and 13 representative lipid traits encompassing eight major lipid classes. These lipid classes include fatty acyls, which comprise palmitic acid [51], stearic acid [51], and caprylic acid [51]; glycerophospholipids, represented by beta-Glycerophosphoric acid [51] and lysophosphatidylinositol [51]; lipoproteins including HDL and LDL [12]; neutral lipids, denoted by triglycerides [12]; medium-chain fatty acids, specifically dodecanoic acid (also known as lauric acid) [51]; steroids and steroid derivatives, encompassing total cholesterol (TC) [12]; and sphingolipids, represented by palmitoyl sphingomyelin [51], 154 SM C16:1 sphingomyelin [52], and 156 SM C18:1 sphingomyelin [52]. 

Detailed information regarding the GWAS data for each lipid trait is presented in Table 1, with additional cohort-specific details available in Appendix A. The GWAS summary data were sourced from popular databases, repositories, and large-scale research consortia. These sources include data for beta-Glycerophosphoric acid (sample size [*N*] = 5912) [51], caprylic acid (*N* = 7802) [51], dodecanoic acid (*N* = 7793) [51], Lysophosphatidylinositol (*N* = 7797) [51], palmitic acid (*N* = 7800) [51], palmitoyl sphingomyelin (*N* = 7814) [51], and stearic acid (*N* = 7803) [51], all of which were obtained from the TwinsUK and KORA cohort (Cooperative Health Research in the Region of Augsburg). Additionally, we accessed lipid GWAS data, including 154 SM C16:1 sphingomyelin C16:1 (*N* = 7428) [13] and 156 SM C18:1 sphingomyelin C18:1 (*N* = 7428) [51] from dataverse, comprising seven cohorts originating from five countries, including the Netherlands, Germany, Australia, Estonia, and the UK. Other lipid GWAS, including LDL (*N* = 188,577) [12], HDL (*N* = 188,577) [12], triglycerides (*N* = 188,577) [12], and TC (*N* = 188,577) [12], were sourced from the Global Lipids Genetics Consortium (GLGC 2013) and the University of Michigan [12]. 

Moreover, we assessed the relationship of seven CAD traits with AD using publicly available GWAS summary data. These traits encompassed angina pectoris (cases = 16,175 and controls = 377,103, *N* = 393,278), cardiac dysrhythmias (cases = 24 681 and controls = 380,919, *N* = 405,600), coronary arteriosclerosis (cases = 20,023 and controls = 377,103, *N* = 397,126), ischemic heart disease (cases = 31,355 and controls = 377,103, *N* = 408,458), myocardial infarction (cases = 11,703 and controls = 377,103, *N* = 388,806), and non-specific chest pain (cases = 31,429 and controls = 377,532, *N* = 408,961). These GWAS summary data comprised full White British samples and were sourced from the Lee Lab for Statistical Genetics (https://www.leelabsg.org/resources, accessed between 1 November 2022 and 30 November 2022) [53]. Some of these data or others from this source have been utilised in previous studies [44,66,67]. Additionally, we utilised GWAS data for CAD from the CARDIoGRAMplusC4D (CGCC), encompassing 22,233 cases and 64,762 controls, with a total sample size of 86,995 [13].

We used two large-scale GWAS summary data for AD in the present study. The first of these comprised clinically diagnosed AD and AD-by-proxy cases (cases = 71,880 and controls = 383,378) [28]. For possible (partial) replication of our findings, we also used AD GWAS summary data from the Informed Genetics Annotated Patient (iGAP) Registry, which includes contributions from research consortia such as the European Alzheimer’s Disease Initiative, Genetic and Environment Risk in Alzheimer’s Disease, Alzheimer’s Disease Genetic Consortium, and Cohorts for Heart and Ageing Research in Genomic Epidemiology) (EADI, GERAD, ADGC, and CHARGE) (N = 17,008 cases and 37,154 controls) [29]. Appendix A provides details about the data sources, along with relevant download links. Comprehensive information on the GWAS datasets and their corresponding quality control protocols can be found in their relevant referenced publications. All GWAS summary data analysed in the present study were derived from individuals of European descent, ensuring genetic homogeneity across our investigations.

### 4.2. Statistical Analyses

We employed well-regarded analytical approaches to comprehensively investigate the genetic relationships of AD with lipids and CAD traits. As represented in Figure 1, we performed analyses at both the SNP and gene levels. SNP-level analysis comprises genetic correlation assessment at the global and local levels. To assess global (genome-wide) and local (specific genomic locations) genetic correlations of AD with lipids and CAD traits, we utilised the linkage disequilibrium score regression (LDSC) [39] and the local analysis of [co]variant association (LAVA) methods [50], respectively. Additionally, at the SNP level, we investigated the potential causal links of AD with lipids and CAD traits using the 2SMR analysis method [40]. Furthermore, we performed analyses at the gene-based level, given that genes are more closely related to biology and can provide greater power for gaining insights into underpinning mechanisms of complex traits. First, we conducted gene-based association analysis and utilised their results to assess genetic overlap at the gene level, complementing our SNP-level genetic overlap assessment [42,43,44,45,46,47,48]. This investigation provides further insights into the interplay of AD with lipids and CAD traits beyond what is possible at the SNP level. Second, we utilised our gene-based association analysis results to assess genes shared by AD, lipids, and CAD traits. 

### 4.3. Assessing Global Genetic Correlation

We conducted a comprehensive genome-wide genetic correlation assessment using the LDSC analysis method [39]. LDSC is a powerful statistical genetic technique employed to estimate the global genetic correlation between multiple traits, utilising GWAS summary data [39]. For our analyses, we used the standalone version of the software (https://github.com/bulik/ldsc, accessed between 20 December 2022 and 25 January 2023). We used precomputed LD scores derived from the European population’s 1000 Genome Project data and used common SNPs in HapMap3 in all our analyses. We first performed pairwise cross-trait analysis, assessing the genetic correlation between AD and each of the 13 lipid traits. Subsequently, we examined the genetic correlation between AD and seven CAD traits. To complete a three-way assessment, we also examined the genetic correlation between all seven CAD traits and the 13 lipid traits.

We employed another set of GWAS summary data for AD to test the reproducibility of our findings. Notably, LDSC adjusts for potential sample overlap between different GWAS datasets when the genetic covariance intercepts are not constrained [39]. In our investigation, we initially performed LDSC analyses without constraining the genetic covariance intercept, probing for any potential sample overlap between AD, CAD, and lipid GWAS datasets. Our findings indicated that the estimated genetic covariance intercepts were not significantly different from zero, signifying no evidence of substantial overlap of samples of AD and lipids or CAD traits, as detailed in Appendix A. Therefore, we reported the LDSC-based genetic correlation results with the genetic covariance intercept constrained. Accounting for multiple pairwise bivariate tests in LDSC (our analyses were performed at a pairwise level), we considered genetic correlation findings statistically significant at ρ < 0.025.

### 4.4. Gene-Level Genetic Overlap Assessment

We assessed the gene-level overlap of AD with lipids and CAD traits, aiming to investigate whether the traits share more genes than would be expected by chance. Gene-level association studies complement SNP-based investigations, providing more interpretable results for deciphering the intricate relationships between AD, lipids, and CAD traits. Our analytical approach in this study is grounded in similar methodologies employed in previous research [42,43,44,45,46,47,48,49]. 

Firstly, we performed gene-based association analysis separately for AD, lipids, and CAD traits using the multi-marker analysis of genomic annotation (MAGMA version 1.08) within the FUMA online platform (https://fuma.ctglab.nl, version 1.5.2, accessed between 4–5 April 2023) [41]. Our objective was to ensure uniform gene-based tests across the board; hence, we restricted our analysis to SNPs shared by AD and each of the respective lipid and CAD trait GWASs. We used the EUR 1000G Phase 3 data as a reference panel in our gene-based association analyses, and SNPs were assigned to genes in MAGMA with a window size of ‘±0 kb’.

Secondly, we leveraged the outcomes of the gene-based analysis for AD, lipids, and CAD traits, extracting genes associated with each trait at a significance threshold of ρ_gene_ < 0.05 for subsequent evaluation of gene-level genetic overlap. At this threshold, we identified the total number of genes linked to AD and the corresponding lipid and CAD trait GWASs. Additionally, we ascertained the total number of genes that exhibited overlap across paired traits. Finally, we assigned lipids and CAD traits as the discovery sets and AD as the target. We used the exact binomial test to compare the expected proportion of gene overlap (the null) with the observed proportion of gene overlap. The anticipated proportion of gene overlap (expected) was calculated as the total number of genes associated with each lipid or CAD trait at ρ_gene_ < 0.05 divided by the total number of genes generated from the gene-based analysis for the respective lipids or CAD traits [42,43]. 

In contrast, the observed proportion of gene overlap represented the ratio of overlapping genes, for example, those shared between AD and CAD-angina pectoris GWAS at ρ_gene_ < 0.05, relative to the total number of genes linked to AD at ρ_gene_ < 0.05 [42,43]. The one-sided exact binomial test, implemented within the R statistical platform, was then employed to evaluate whether the observed proportion of the gene overlap surpassed what would be expected by random chance. For a gene-level genetic overlap to be deemed significant, the observed proportion of gene overlap must exhibit statistical significance beyond the null hypothesis. In the ‘result explained’ sub-section (under Table 4), we provided additional information on the analyses, using an example from our findings.

### 4.5. Identifying Genes Shared by AD, Lipids, and CAD Traits

We assess genes shared by AD across lipids and CAD traits at two levels. First, we identify GWS sentinel genes (P_gene_ < 2.60 × 10^−6^), genes that were already GWS in our data and overlap between AD and each of the lipids or CAD traits assessed. Further, we identify GWS sentinel genes overlapping AD and two or more lipids or CAD traits for insights into genes shared across these disorders. Second, using a similar approach to previous studies, we employed the FCP method to merge gene association *p* values for AD and each lipid and CAD trait, allowing us to identify their shared genes that achieved GWS. For this analysis, we used the results from equivalent gene-based association analyses (outputs from MAGMA) for the pair of AD GWASs and each lipid or CAD trait. We first identified GWS genes for AD and each lipid or CAD trait separately, applying an adjusted *p* value of 2.60 × 10^−6^. We then combined the association *p* values using the FCP approach. The FCP analysis results helped us identify shared genes in two broad categories. First are the GWS genes for AD shared by lipids or CAD traits and vice versa (with evidence for improvement in the FCP analysis). Second are the genes not GWS in either AD or lipids or CAD traits GWAS but reaching this status after the FCP analysis (putatively novel shared genes based on our data).

### 4.6. Causal Relationship Assessment 

We conducted 2SMR analyses to comprehensively investigate the bi-directional causal relationship between CAD traits, lipids, and AD. Inclusion criteria for the 2SMR analysis encompassed only those traits that exhibited significant global genetic correlation and gene-level overlap with AD. MR is a robust statistical method that leverages instrumental variables to mimic randomised control trials, providing a cost-effective means of estimating causality between two traits, referred to as the ‘exposure’ and ‘outcome’ variables [40]. In this study, our 2SMR analysis was performed using the standalone 2SMR package (https://mrcieu.github.io/TwoSampleMR/) (accessed between 23 October 2023 and 31 January 2024), implemented within the R statistical package (v4.3.0).

For our analysis, we assessed lipids (HDL, LDL, triglycerides, and total cholesterol) and the seven CAD traits’ GWAS as the ‘exposure’ against the AD GWAS (from the Jansen et al. dataset [28]) as the ‘outcome’ variables. Instrumental variables (IVs) for the exposure data were selected at the genome-wide significance level (ρ < 5 × 10^−8^), and LD clumping was performed (at r^2^ < 0.001) to guarantee the independence of the IVs. Subsequently, we extracted the IVs from the outcome data and carried out data harmonisation to ensure that SNP effects corresponded to the same allele for both the exposure and outcome data. In a bi-directional analysis, we also performed a 2SMR analysis using AD GWAS as the exposure and both lipids and CAD traits as outcome data. Moreover, we used data from another AD GWAS (containing only clinically-diagnosed cases) to test the reproducibility of our 2SMR findings [29].

Our primary 2SMR approach employed the inverse variance-weighted (IVW) model, recognised as the most powerful model for detecting causal associations in MR analysis when all instruments are valid [40]. To evaluate the potential violation of the assumption of no horizontal or directional pleiotropy, we employed the MR-Egger intercept, expecting that the intercept should not significantly deviate from zero where the assumption of no unbalanced pleiotropy holds. Additionally, we implemented two supplementary MR models, namely the weighted-median and the MR-Egger regression methods, for sensitivity testing and to complement the IVW model in assessing causality [68,69]. The weighted-median and MR-Egger models are robust to genetic heterogeneity. They can provide estimates even when a substantial proportion of the IVs are invalid, making them essential for rigorous causal assessment. Lastly, we implemented the Mendelian randomisation pleiotropy residual sum and outlier (MR-PRESSO) method to detect (global test) and correct (outlier test) horizontal pleiotropy and test for significant distortion in the causal estimates [70].

Additionally, Figure 4 outlines the principles of Mendelian randomisation analysis and assumptions that need to be met for unbiased causal estimates between all seven CAD traits and four lipids (HDL, LDL, triglycerides, and total cholesterol) and AD [50,70].

### 4.7. Local Genetic Correlation Analysis

To enhance our understanding beyond the broad global genetic correlations, we conducted local genetic correlation analyses for the relationship of AD with lipids and CAD traits. Global genetic correlations, such as those estimated by the LDSC method, provide valuable insights into the average genetic relationships between two traits across the entire genome [39,72]. However, they may mask significant genetic correlations when local genetic effects act in opposing directions at different genetic loci. To refine our comprehension of genetic relationships, we harnessed the power of a recently developed integrated framework for local genetic correlation analysis known as LAVA (local analysis of [co]variant association) [50].

LAVA is a versatile method that can simultaneously model multiple binary or continuous phenotypes, facilitating univariate, bivariate, and multivariate assessments between phenotypes, including partial correlation analysis between two or more traits [50]. We implemented LAVA through RStudio, enabling us to estimate bivariate local genetic correlations for AD, lipids (specifically HDL, LDL, triglycerides, and total cholesterol), and seven CAD traits. Our analysis first evaluated potential sample overlap between the included traits, which was assessed using the LDSC approach [39]. To ensure the integrity of LAVA results, aligning the direction of effect allele consistently across all phenotypes is vital. LAVA achieves this by extracting SNPs common to the GWAS summary statistics data and harmonising their effect alleles with reference genotype data (1000G EUR) before analysis [50]. We utilised the semi-LD independent locus definition file provided by the program developer in the present study [50].

Subsequently, we conducted LAVA-based univariate analyses to estimate local genetic heritability for AD, lipids, and CAD traits. A robust signal in the local univariate analysis is a prerequisite for detecting bivariate genetic correlations in LAVA [50]. From the outcomes of the univariate analysis, we selected traits for bivariate analysis at a significance level of ρ < 5 × 10^−2^ (to allow for more bivariate analysis without the risk of false positives as recently implemented in [49]). Finally, similar to a recent study [49], we conducted pairwise bivariate local genetic correlation analyses across the genome for AD, lipids (HDL, LDL, triglycerides, and total cholesterol), and seven CAD traits. Applying a Bonferroni correction for the highest number of bivariate tests performed (AD Jansen et al. [28] vs. CAD (35 bivariate tests), we considered bivariate analysis results significant at ρ < 1.4 × 10^−3^ (that is 0.05/35). 

## 5. Conclusions

Our study contributes to the ongoing investigation into the complex relationships between AD, lipids, and CAD traits. Despite the previous literature suggesting connections of AD with lipids and CAD, the precise nature of these relationships remained unresolved. Through a comprehensive analysis integrating genetic data from large cohorts, we systematically assessed the cross-trait genetic overlap of AD with 13 representative lipids and seven CAD traits, leveraging robust analytical methods. Our study uncovered several key findings. The global genetic correlation analyses revealed significant associations between AD and specific lipid traits, including LDL, triglycerides, and total cholesterol, suggesting shared genetic components that may predispose individuals to AD and dyslipidaemia. Furthermore, we identified a positive genetic correlation between AD and various CAD traits, indicating a potential comorbid relationship between these disorders. Gene-level analyses largely reinforce these findings and highlight the genetic overlap between AD and high-density lipoproteins (HDLs), which was not evident in the global genetic correlation assessment. Notably, our bi-directional Mendelian randomisation analyses did not provide evidence for a causal link between AD, lipids, and CAD traits, underscoring the complexity of these genetic relationships and indicating that shared genetic susceptibility may better explain their observed correlations. Local genetic correlation analysis using the LAVA method pinpointed specific genomic loci, particularly on chromosomes 6, 8, 17, and 19, significantly contributing to the association between AD and these traits, agreeing largely with our identified shared genes for AD and each lipid or CAD trait. Completing a three-way analysis, we confirm a strong genetic correlation between lipids and CAD traits, with HDL and sphingomyelin demonstrating negative correlations. Current findings provide valuable insights into the genetic underpinnings of AD and its relationship with lipids and CAD traits, offering potential avenues for further research and therapeutic development. Overall, our study enhances understanding of the complex genetic landscape of AD and its connections with cardiovascular health, providing a foundation for further investigations aimed at unravelling the underlying mechanisms and identifying novel treatment strategies.

## Figures and Tables

**Figure 1 ijms-25-08814-f001:**
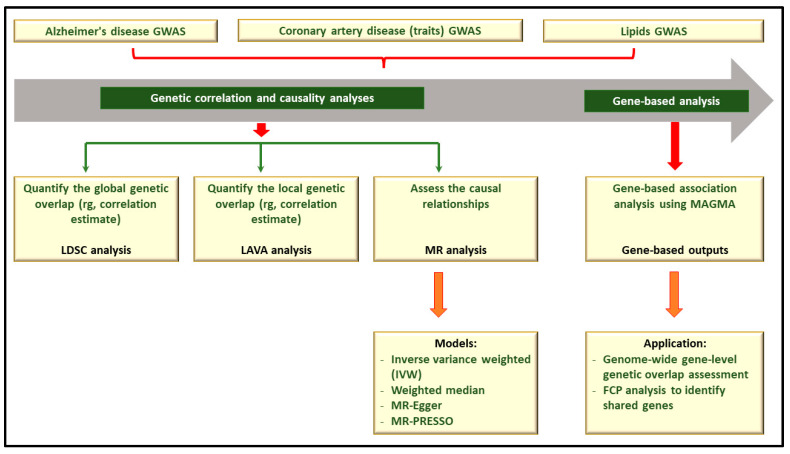
Study design and workflow: assessing shared genetic associations between AD, lipids, and CAD traits. FCP: Fisher’s combined *p* value; MAGMA–multi-marker analysis of genomic annotation; GWAS: genome-wide association studies; LDSC: linkage disequilibrium score regression; LAVA: local analysis of [co]variant association; MR: Mendelian randomisation; MR-PRESSO: Mendelian randomisation pleiotropy residual sum and outlier.

**Figure 2 ijms-25-08814-f002:**
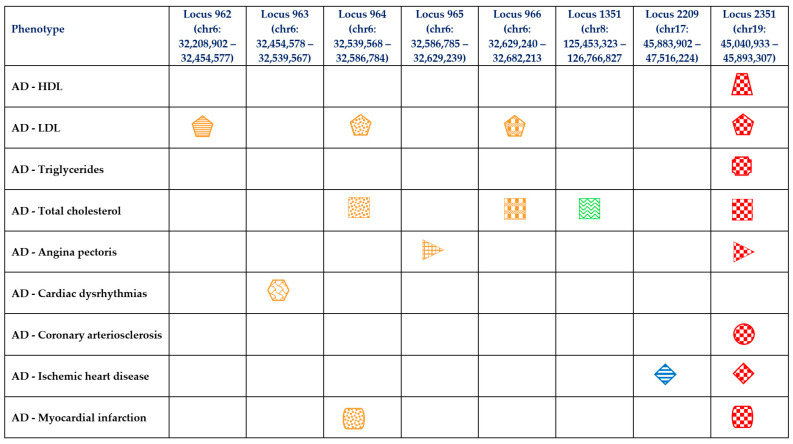
Loci contributing disproportionately to the genetic correlation of Alzheimer’s disease with lipids and CAD traits. AD: Alzheimer’s disease; CAD: coronary artery disease; Chr: chromosome; HDL: high-density lipoprotein; LDL: low-density lipoprotein. Representative symbol for phenotype—HDL: 
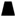
; LDL: 
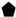
; Triglycerides: 
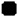
; Total cholesterol: 
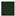
; Angina pectoris: 
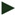
; Cardiac dysrhythmias: 
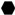
; Coronary arteriosclerosis: 
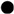
; Ischemic heart disease: 
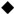
; Myocardial infarction: 
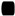
. Representative colour for chromosome—Chromosome 6: ORANGE; Chromosome 8: GREEN; Chromosome 17: BLUE; Chromosome 19: RED. Representative pattern fill for locus position—962: 
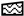
; 963: 
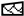
; 964: 
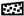
; 965: 
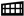
; 966: 
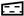
; 1351: 
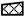
; 2209: 
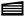
; 2351: 
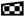
.

**Figure 3 ijms-25-08814-f003:**
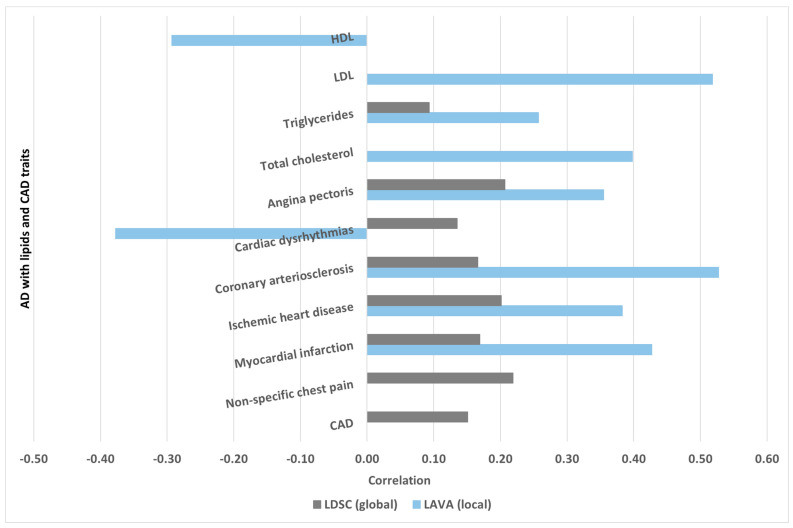
Comparison of LDSC and LAVA assessments for lipids, CAD traits, and Alzheimer’s disease. Figure explained: Figure 3 illustrates the comparative results of genetic correlations between various traits assessed using two distinct methods: LDSC and LAVA. This figure provides a visual representation of how these methods yield different insights into the genetic relationships of AD with lipids and CAD traits. For example, the figure highlights significant genetic correlations detected between pairs of traits using LDSC and LAVA and shows where the two methods agree or differ. By analysing these differences, Figure 3 underscores the importance of using multiple analytical approaches to gain a comprehensive understanding of genetic correlations, offering insights into how genetic factors may differentially influence the studied traits. There are three areas of comparison: (a) Significant in LDSC but not in LAVA: (i) AD and non-specific chest pain: a significant positive correlation in LDSC but only marginal significance in LAVA; (ii) AD and CAD: a significant positive correlation in LDSC but only nominally significant in LAVA. (b) Significant in LAVA but not in LDSC: (i) AD and HDL: significant negative correlation in LAVA but not significant in LDSC; (ii) AD and LDL: significant positive correlation in LAVA, but not significant in LDSC; (iii) AD and TC: significant positive correlation in LAVA, but not significant in LDSC. (c) Concordance in effect for significant results: (i) AD and cardiac dysrhythmias: significant positive correlation in LDSC but a significant negative correlation in LAVA. AD: Alzheimer’s disease; CAD traits: coronary artery disease traits; HDL: high-density lipoprotein; TC: total cholesterol; LAVA: local analysis of [co]variant associations; LDL: low-density lipoprotein; LDSC: linkage disequilibrium score regression.

**Figure 4 ijms-25-08814-f004:**
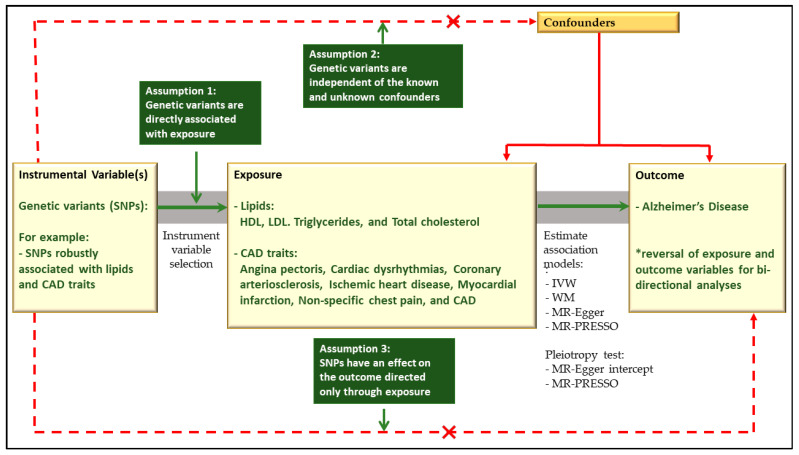
Two-sample Mendelian randomisation assumptions and workflow summary [40,71]. A causal effect of exposure can be inferred only if the three assumptions for the instrumental variables are met [40,71]. The instrumental variable must have a robust association with the exposure variable (Assumption 1), the instrumental variable may not be related to any known or unknown confounding causal factors of the exposure variable and the outcome variable (Assumption 2), and the instrumental variable may only be related to the outcome variable through the exposure variable [40,71]. Assumptions 2 and 3 can help determine the absence of pleiotropy for the instrumental variable [40,71]. CAD traits: coronary artery disease traits; HDL: high-density lipoprotein; LDL: low-density lipoprotein; MR: two-sample Mendelian randomisation; MR-PRESSO: Mendelian randomisation pleiotropy residual sum and outlier; SNPs: single-nucleotide polymorphisms. * We conducted a bidirectional MR analysis by changing the direction of the analysis. Lipids and CAD traits were used as exposure, and AD was used as the outcome variable. In the reverse analysis, we used AD as the exposure, while lipids and CAD traits were utilised as the outcome variables.

**Table 1 ijms-25-08814-t001:** Summary of the GWAS data analysed.

GWAS Summary Statistics	Cases	Controls	Sample Size	Ancestry	Phenotype Source/Definition
AD				European	
Main (Jansen et al. [28])	71,880	383,378	455,258		Clinically diagnosed and UKB AD-by-proxy2
Validation (Lambert et al. [29]) *	17,008	37,154	54,162		Data from the EADI, GERAD, ADGC, and CHARGE study
LIPID				European	
Sphingolipids:					
Palmitoyl sphingomyelin (Shin et al. [51])			7814		Data from the TwinsUK and KORA study
154 SM C16:1 sphingomyelin (Draisma et al. [52])			7428		Data from dataverse
156 SM C18:1 sphingomyelin (Draisma et al. [52])			7428		Data from dataverse
Glycerophospholipids:					
Beta-glycerophosphoric acid (Shin et al. [51])			5912		Data from the TwinsUK and KORA study
Lysophosphatidylinositol (Shin et al. [51])			7797		Data from the TwinsUK and KORA study
Fatty Acyls:					
Palmitic acid (Shin et al. [51])			7800		Data from the TwinsUK and KORA study
Stearic acid (Shin et al. [51])			7803		Data from the TwinsUK and KORA study
Fatty Acyls [lipids or lipid-like molecules]:					
Caprylic acid (Shin et al. [51])			7802		Data from the TwinsUK and KORA study
Organic compounds known as medium-chain fatty acids:					
Dodecanoic acid (Shin et al. [51]), (also known as lauric acid)			7793		Data from the TwinsUK and KORA study
Lipoproteins:					
HDL (GLGC [12])			188,577		Data from the GLGC
LDL (GLGC [12])			188,577		Data from the GLGC
Neutral lipids:					
TG (GLGC [12])			188,577		Data from the GLGC
Steroids and steroid derivatives:					
TC (GLGC [12])			188,577		Data from the GLGC
CAD trait				European	
Angina pectoris Phecode 411.3 (Lee Lab [53])	16,175	377,103	393,278		Full European data subset from the Lee Lab
Cardiac dysrhythmias Phecode 427 (Lee Lab [53])	24,681	380,919	405,600		Full European data subset from the Lee Lab
Coronary atherosclerosis Phecode 411.4 (Lee Lab [53])	20,023	377,103	397,126		Full European data subset from the Lee Lab
Ischemic heart disease Phecode 411 (Lee Lab [53])	31,355	377,103	408,458		Full European data subset from the Lee Lab
Myocardial infarction Phecode 411.2 (Lee Lab [53])	11,703	377,103	388,806		Full European data subset from the Lee Lab
Non-specific chest pain Phecode 418 (Lee Lab [53])	31,429	377,532	408,961		Full European data subset from the Lee Lab
CARDIoGRAMplusC4D (CGCC [12])	22,233	64,762	86,995		Data from the CGCC

The ‘clinically diagnosed AD’ combined data from three case-control cohorts. The “AD-by proxy” data were derived from the UKB phenotype definition, identifying individuals with biological parents affected by AD. Information on the parents’ current age and, when applicable, age at death was included alongside the GWAS data. A substantial genetic correlation of 0.81 exists between “clinically diagnosed AD” and “AD-by proxy”, supporting their combination, as elaborated further in the related publication [28]. AD: Alzheimer’s disease; ADGC: Alzheimer’s disease genetic consortium; CAD: Coronary artery disease; CGCC: CARDIoGRAMplusC4D consortium; CHARGE: Cohorts for heart and ageing research in genomic epidemiology; EADI: European Alzheimer’s disease initiative; GERAD: Genetic and environment risk in Alzheimer’s disease; GLGC: Global lipids genetics consortium; HDL: High-density lipoprotein; KORA: Cooperative Health Research in the Region Augsburg; LDL: Low-density lipoprotein; TC: Total cholesterol; TG: Triglycerides; UKB: United Kingdom Biobank. * The validation set data were used for reproducibility testing in LDSC, gene-based, MR, and LAVA analysis. We note the data are not completely independent; hence, we use ‘partial replication’ as is appropriate.

**Table 2 ijms-25-08814-t002:** Results of global genetic correlation analysis of Alzheimer’s disease with lipids and CAD traits using LDSC.

**AD**	**Lipids**	**rg**	**Se**	** *p* **
	Palmitoyl sphingomyelin	−0.03	4.22 × 10^−2^	4.96 × 10^−1^
	154 SM C16:1 sphingomyelin	−0.02	1.07 × 10^−1^	8.78 × 10^−1^
	156 SM C18:1 sphingomyelin	0.14	1.37 × 10^−1^	3.10 × 10^−1^
	beta-Glycerophosphoric acid	0.05	5.69 × 10^−2^	4.28 × 10^−1^
	Lysophosphatidylinositol	0.04	5.48 × 10^−2^	5.11 × 10^−1^
	Palmitic acid	0.00	5.08 × 10^−2^	9.96 × 10^−1^
AD	Stearic acid	−0.03	4.34 × 10^−2^	5.49 × 10^−1^
	Caprylic acid	0.00	4.56 × 10^−2^	9.51 × 10^−1^
	Dodecanoic acid	−0.01	3.94 × 10^−2^	7.20 × 10^−1^
	HDL	−0.05	3.85 × 10^−2^	1.81 × 10^−1^
	LDL	0.13	7.10 × 10^−2^	6.49 × 10^−2^
	TG	0.09	3.64 × 10^−2^	1.09 × 10^−2^
	TC	0.13	7.01 × 10^−2^	5.48 × 10^−2^
AD	CAD traits	Rg	Se	*p*
	Angina pectoris	0.21	3.55 × 10^−2^	5.88 × 10^−9^
	Cardiac dysrhythmias	0.14	3.78 × 10^−2^	3.49 × 10^−4^
	Coronary arteriosclerosis	0.17	2.96 × 10^−2^	2.26 × 10^−8^
AD	Ischemic heart disease	0.20	3.13 × 10^−2^	1.39 × 10^−10^
	Myocardial infarction	0.17	3.84 × 10^−2^	1.03 × 10^−5^
	Non-specific chest pain	0.22	3.91 × 10^−2^	2.06 × 10^−8^
	CAD	0.15	4.25 × 10^−2^	3.74 × 10^−4^

AD: Alzheimer’s disease, data from Jansen et al. [28]; CAD traits: coronary artery disease traits [12,53]; HDL: high-density lipoprotein; LDL: low-density lipoprotein; TC: total cholesterol; TG: triglycerides; *p*: estimated ρ value; rg: global genetic correlation estimates; se: standard error; LDSC: linkage disequilibrium score regression.

**Table 3 ijms-25-08814-t003:** Results of global genetic correlation analysis between CAD traits and lipids using LDSC.

CAD Trait	Lipids Trait	rg	Se	*p*
Angina pectoris	HDL	−0.39	4.77 × 10^−2^	1.55 × 10^−16^
	LDL	0.28	3.66 × 10^−2^	5.68 × 10^−14^
	TG	0.41	5.73 × 10^−2^	8.92 × 10^−13^
	TC	0.23	3.44 × 10^−2^	1.07 × 10^−11^
Cardiac dysrhythmias	HDL	−0.18	3.60 × 10^−2^	2.93 × 10^−7^
	TG	0.14	4.00 × 10^−2^	3.70 × 10^−4^
Coronary arteriosclerosis	HDL	−0.36	4.44 × 10^−2^	8.72 × 10^−16^
	LDL	0.3	3.78 × 10^−2^	4.93 × 10^−15^
	TG	0.37	4.62 × 10^−2^	2.11 × 10^−15^
	TC	0.25	3.59 × 10^−2^	2.10 × 10^−12^
	154 SM C16:1 sphingomyelin	−0.31	1.23 × 10^−1^	1.18 × 10^−2^
Ischemic heart disease	HDL	−0.38	4.66 × 10^−2^	2.65 × 10^−16^
	LDL	0.28	3.55 × 10^−2^	3.08 × 10^−15^
	TG	0.4	5.07 × 10^−2^	2.99 × 10^−15^
	TC	0.24	3.28 × 10^−2^	1.73 × 10^−13^
Myocardial infarction	HDL	−0.37	5.27 × 10^−2^	1.25 × 10^−12^
	LDL	0.29	3.82 × 10^−2^	6.62 × 10^−14^
	TG	0.41	5.65 × 10^−2^	2.82 × 10^−13^
	TC	0.26	3.59 × 10^−2^	3.98 × 10^−13^
Non-specific chest pain	HDL	−0.32	4.40 × 10^−2^	5.38 × 10^−13^
	LDL	0.14	3.35 × 10^−2^	2.86 × 10^−5^
	TG	0.31	5.29 × 10^−2^	7.10 × 10^−9^
	TC	0.1	3.11 × 10^−2^	1.96 × 10^−3^
CAD	HDL	−0.37	4.40 × 10^−2^	6.52 × 10^−17^
	LDL	0.39	4.52 × 10^−2^	1.48 × 10^−17^
	TG	0.42	4.39 × 10^−2^	1.41 × 10^−21^
	TC	0.35	4.24 × 10^−2^	2.67 × 10^−16^

CAD: coronary artery disease traits; HDL: high-density lipoprotein; LDL: low-density lipoprotein; TC: total cholesterol; TG: triglycerides; *p*: estimated ρ value; rg: global genetic correlation estimates; se: standard error; LDSC: linkage disequilibrium score regression.

**Table 4 ijms-25-08814-t004:** Results of gene-level overlap assessment of Alzheimer’s disease with lipids and CAD traits.

Discovery Set			Target Set			Number of Genes	Proportion of Gene Overlap		Binomial Test
Lipids and CAD Traits	Total Number of Genes in the Discovery Set (Lipid or CAD Trait)	Number of Genes in the Discovery Set P_gene_ < 0.05	AD	Total Number of Genes in the Target Set (AD)	Number of Genes in the Target Set at P_gene_ < 0.05	Overlapping the Discovery and the Target Sets at P_gene_ < 0.05	Expected (%)	Observed (%)	*p* Value
* HDL	17,683	1880	AD	17,683	1768	294	10.6	16.6	9.84 × 10^−15^
LDL	17,669	1766	AD	17,669	1769	267	10.0	15.1	1.28 × 10^−11^
Triglycerides	17,671	1743	AD	17,671	1769	273	9.9	15.4	2.24 × 10^−13^
Total cholesterol	17,683	1988	AD	17,683	1767	320	11.2	18.1	2.20 × 10^−16^
Angina pectoris	18,960	2175	AD	18,960	1843	260	11.5	14.1	3.65 × 10^−4^
Cardiac dysrhythmias	18,960	1776	AD	18,960	1843	212	9.4	11.5	1.48 × 10^−3^
Coronary arteriosclerosis	18,960	2524	AD	18,960	1843	333	13.3	18.1	4.73 × 10^−9^
Ischemic heart disease	18,960	2710	AD	18,960	1843	315	14.3	17.1	4.60 × 10^−4^
Myocardial infarction	18,960	1995	AD	18,960	1843	244	10.5	13.2	1.18 × 10^−4^
Non-specific chest pain	18,960	1943	AD	18,960	1843	212	10.2	11.5	3.69 × 10^−2^
CAD	17,735	1601	AD	17,735	1781	201	9.0	11.3	6.26 × 10^−4^

AD: Alzheimer’s disease; CAD: coronary artery disease; HDL: high-density lipoprotein; LDL: low-density lipoprotein; *p*: *p*-value. * Result explained (using AD–HDL as an example): comparison of the expected proportions of gene overlap (null hypothesis) with the observed proportion of gene overlap. The expected proportion of gene overlap equals the number of genes associated with the discovery set (lipids and CAD traits) at P_gene_ < 0.05 divided by the total number of genes for each respective discovery set. For example, for Lipid—HDL, the number of genes associated with the discovery set at P_gene_ < 0.05 (1880)/the total number of genes (17,683). The observed proportion of genes equals the number of overlapping genes divided by the total number of genes associated with the target set. Using HDL–AD analysis as an example, the number of overlapping genes at P_gene_ < 0.05 = 294/the total number of genes associated with the target set at P_gene_ < 0.05 (1768). To test whether the observed proportion of genes is more than expected by chance, we performed a one-sided exact binomial test within the R statistical platform [binom.test(294,1768,0.106,alternative = c(“greater”))].

**Table 5 ijms-25-08814-t005:** Genome-wide significant genes overlapping AD, lipids, and CAD traits.

Genes	Chr	START (hg19)	STOP (hg19)	AD, Lipids, and CAD Traits
GWS genes (sentinel) overlapping AD and two or more CAD or lipid traits				
APOC1	19	45,417,504	45,422,606	AD, CA, IHD, MI
APOC4	19	45,445,495	45,452,820	AD, HDL, LDL, TC
APOC4-APOC2	19	45,445,495	45,452,822	AD, HDL, TC
APOE	19	45,409,011	45,412,650	AD, AP, CA, HDL, IHD, LDL, MI, TC, TG
BCL3	19	45,250,962	45,263,301	AD, LDL, TC
CBLC	19	45,281,126	45,303,891	AD, LDL, TC
CEACAM19	19	45,165,545	45,187,631	AD, LDL, TC
IGSF23	19	45,116,940	45,140,081	AD, LDL, TC
NKPD1	19	45,653,008	45,663,408	AD, LDL, TC
PVR	19	45,147,098	45,166,850	AD, LDL, TC
PVRL2	19	45,349,432	45,392,485	AD, CA, IHD, LDL, TG, TC
TOMM40	19	45,393,826	45,406,946	AD, CA, HDL, IHD, LDL, MI, TC, TG
ZNF652	17	47,366,568	47,439,835	AD, AP, CA, IHD
Genes reaching GWS in the FCP analysis overlapping AD and two or more CAD or lipid traits				
ACMSD	2	135,596,117	135,659,604	AD, LDL, TC
ICA1L	2	203,640,690	203,736,708	AD, AP, LDL, TC
WDR12	2	203,739,505	203,879,521	AD, AP, LDL, TC
CARF	2	203,776,937	203,851,786	AD, AP, LDL, TC
PRRC2A	6	31,588,497	31,605,548	AD, CA, IHD
BAG6	6	31,606,805	31,620,482	AD, CA, IHD, NSCP
C6orf10	6	32,256,303	32,339,684	AD, CA, IHD, TC
HLA-DRA	6	32,407,619	32,412,823	AD, AP, CA, IHD, MI, HDL, TG
HLA-DQB1	6	32,627,244	32,636,160	AD, AP, CA, LDL
TMEM106B	7	12,250,867	12,282,993	AD, AP, IHD
NDUFAF6	8	95,907,995	96,128,683	AD, CA, IHD
TRIB1	8	126,442,563	126,450,647	AD, CA, HDL
DOC2A	16	30,016,830	30,034,591	AD, CD, IHD
ZNF668	16	31,072,164	31,085,641	AD, LDL, TC
PRSS8	16	31,142,756	31,147,083	AD, LDL, TC
PLCG2	16	81,772,702	81,991,899	AD, AP, CA
RP11-81K2.1	17	47,448,102	47,554,350	AD, MI, NSCP
PHB	17	47,481,414	47,492,246	AD, CD, MI, NSCP
APOC2	19	45,449,243	45,452,822	AD, LDL, TG
RSPH6A	19	46,298,968	46,318,577	AD, LDL, TC

AD: Alzheimer’s disease, AP: angina pectoris, CA: coronary arteriosclerosis, CAD: coronary artery disease, CD: cardiac dysrhythmias, IHD: ischemic heart disease, MI: myocardial infarction, HDL: high-density lipoprotein, LDL: low-density lipoprotein, NSCP: non-specific chest pain, TG: triglycerides, TC: total cholesterol, Chr: chromosome, FCP: Fisher’s combined *p* value, GWS: genome-wide significant, hg: human genome build.

**Table 6 ijms-25-08814-t006:** Results of bi-directional Mendelian randomisation analyses of Alzheimer’s disease with lipids and CAD traits.

Outcome	Exposure	nIV	IVW		Weighted Median		MR-Egger		* P_pleiotropy_	# P_heterogeneity_	MR-PRESSO			
			OR (95% CI)	*p* Value	OR (95% CI)	*p* Value	OR (95% CI)	*p* Value			RAW OR (95% CI)	*p* Value	Corrected OR (95% CI)	*p* Value
	HDL	3	1.02 (0.94–1.12)	5.91 × 10^−1^	1.04 (0.94–1.14)	4.56 × 10^−1^	1.15 (0.79–1.67)	6.01 × 10^−1^	6.49 × 10^−1^	7.24 × 10^−1^	-	-	-	-
	LDL	62	1.00 (0.99–1.02)	4.19 × 10^−1^	1.00 (0.99–1.02)	7.19 × 10^−1^	1.00 (0.99–1.02)	6.19 × 10^−1^	9.93 × 10^−1^	4.15 × 10^−1^	1.00 (0.99–1.02)	4.23 × 10^−1^	NA	NA
	Triglycerides	43	1.01 (0.99–1.02)	2.13 × 10^−1^	1.00 (0.99–1.03)	5.74 × 10^−1^	1.00 (0.98–1.02)	9.33 × 10^−1^	2.31 × 10^−1^	5.84 × 10^−1^	1.01 (1.00–1.02)	2.06 × 10^−1^	NA	NA
	Total cholesterol	66	1.00 (0.99–1.02)	4.65 × 10^−1^	1.00 (0.98–1.02)	7.41 × 10^−1^	1.00 (0.98–1.02)	9.50 × 10^−1^	6.93 × 10^−1^	4.92 × 10^−1^	1.00 (0.99–1.02)	4.66 × 10^−1^	NA	NA
	Angina pectoris	15	1.00 (0.99–1.02)	4.70 × 10^−1^	1.01 (0.99–1.03)	2.53 × 10^−1^	1.01 (0.98–1.04)	5.09 × 10^−1^	7.05 × 10^−1^	1.44 × 10^−1^	1.00 (0.99–1.02)	4.81 × 10^−1^	NA	NA
AD	Cardiac dysrhythmias	24	1.00 (0.99–1.02)	4.15 × 10^−1^	1.01 (0.99–1.02)	3.95 × 10^−1^	1.02 (1.00–1.05)	1.13 × 10^−1^	1.64 × 10^−1^	9.40 × 10^−1^	1.00 (1.00–1.01)	2.99 × 10^−1^	NA	NA
	Coronary arteriosclerosis	37	1.00 (1.00–1.01)	4.20 × 10^−1^	1.01 (0.99–1.02)	3.53 × 10^−1^	1.02 (1.00–1.04)	3.06 × 10^−2^	4.16 × 10^−2^	3.88 × 10^−1^	1.00 (1.00–1.01)	4.25 × 10^−1^	NA	NA
	Ischemic heart disease	33	1.01 (1.00–1.02)	7.62 × 10^−2^	1.01 (0.99–1.03)	2.15 × 10^−1^	1.01 (0.98–1.03)	5.06 × 10^−1^	8.57 × 10^−1^	2.75 × 10^−1^	1.01 (1.00–1.02)	8.58 × 10^−2^	NA	NA
	Myocardial infarction	14	1.00 (0.99–1.02)	5.97 × 10^−1^	0.99 (0.98–1.01)	4.89 × 10^−1^	1.02 (0.99–1.04)	2.68 × 10^−1^	3.22 × 10^−1^	2.24 × 10^−1^	1.00 (0.99–1.02)	6.06 × 10^−1^	NA	NA
	Non-specific chest pain	1	-	-	-	-	-	-	-	-	-	-	-	-
	CAD	13	1.00 (0.99–1.01)	9.41 × 10^−1^	1.00 (0.98–1.01)	7.21 × 10^−1^	0.99 (0.95–1.03)	5.71 × 10^−1^	5.70 × 10^−1^	9.27 × 10^−1^	1.00 (0.99–1.01)	9.06 × 10^−1^	NA	NA
HDL		2	0.77 (0.30–2.00)	5.86 × 10^−1^	-	-	-	-	-	1.26 × 10^−1^	-	-	-	-
LDL		10	1.00 (0.79–1.26)	9.81 × 10^−1^	1.14 (0.82–1.57)	4.33 × 10^−1^	1.03 (0.33–3.22)	9.56 × 10^−1^	9.51 × 10^−1^	6.00 × 10^−1^	1.00 (0.80–1.24)	9.79 × 10^−1^	NA	NA
Triglycerides		10	0.90 (0.69–1.18)	4.42 × 10^−1^	0.75 (0.54–1.04)	8.74 × 10^−2^	1.93 (0.56–6.72)	3.31 × 10^−1^	2.55 × 10^−1^	1.19 × 10^−1^	0.90 (0.69–1.18)	4.62 × 10^−1^	NA	NA
Total cholesterol		10	0.94 (0.74–1.18)	5.85 × 10^−1^	1.09 (0.80–1.47)	5.97 × 10^−1^	0.89 (0.29–2.77)	8.46 × 10^−1^	9.30 × 10^−1^	4.99 × 10^−1^	0.94 (0.75–1.17)	5.85 × 10^−1^	NA	NA
Angina pectoris		23	0.85 (0.65–1.11)	2.25 × 10^−1^	1.17 (0.81–1.67)	4.07 × 10^−1^	1.43 (0.93–2.20)	1.14 × 10^−1^	8.07 × 10^−3^	2.66 × 10^−1^	0.85 (0.65–1.11)	2.37 × 10^−1^	NA	NA
Cardiac dysrhythmias	AD	27	0.96 (0.84–1.10)	5.50 × 10^−1^	1.09 (0.90–1.31)	3.85 × 10^−1^	1.11 (0.93–1.32)	2.76 × 10^−1^	2.62 × 10^−2^	8.29 × 10^−1^	0.96 (0.86–1.08)	4.92 × 10^−1^	NA	NA
Coronary arteriosclerosis		22	0.96 (0.75–1.22)	7.29 × 10^−1^	1.32 (0.96–1.82)	9.06 × 10^−2^	1.23 (0.82–1.85)	3.36 × 10^−1^	1.59 × 10^−1^	2.96 × 10^−1^	0.96 (0.75–1.22)	7.32 × 10^−1^	NA	NA
Ischemic heart disease		2	1.03 (0.40–2.61)	9.57 × 10^−1^	-	-	-	-	-	3.06 × 10^−1^	-	-	-	-
Myocardial infarction		24	1.09 (0.82–1.44)	5.61 × 10^−1^	1.19 (0.78–1.79)	4.21 × 10^−1^	1.23 (0.75–2.01)	4.27 × 10^−1^	5.67 × 10^−1^	8.77 × 10^−1^	1.09 (0.86–1.37)	4.85 × 10^−1^	NA	NA
Non-specific chest pain		25	0.94 (0.81–1.89)	3.87 × 10^−1^	0.98 (0.80–1.21)	8.61 × 10^−1^	0.92 (0.74–1.15)	4.60 × 10^−1^	8.20 × 10^−1^	8.16 × 10^−1^	0.94 (0.82–1.06)	3.24 × 10^−1^	NA	NA
CAD		9	1.50 (0.73–3.10)	2.72 × 10^−1^	2.14 (0.83–5.47)	1.11 × 10^−1^	2.03 (0.08–53.19)	6.82 × 10^−1^	8.57 × 10^−1^	6.12 × 10^−1^	1.50 (0.79–2.86)	2.51 × 10^−1^	NA	NA

AD: Alzheimer’s disease; CAD traits: coronary artery disease traits; HDL: high-density lipoprotein; IVW: inverse variance weighted; LDL: low-density lipoprotein; MR-Egger: Mendelian randomisation-Egger; MR-PRESSO: Mendelian randomisation pleiotropy residual sum and outlier; nIV: number of instrumental variables (nSNPs: number of single-nucleotide polymorphisms); OR: odds ratio, 95% CI: 95% confidence interval. * MR-Egger intercepts *p*-value, assessing potential horizontal or directional pleiotropy. # *p*-value for the heterogeneity test. Note: spaces marked ‘NA’ indicate no outlier-corrected results (potential pleiotropy not detected) in the MR-PRESSO analyses, while those with a dash (-) represent no results were produced due to insufficient instruments.

**Table 7 ijms-25-08814-t007:** Results of bivariate local genetic correlation of Alzheimer’s disease with lipids and CAD traits using LAVA.

Locus	Chr	Start	Stop	SNP (n)	Phenotype1	Phenotype2	RHO	R2	*p*	Mean.RHO
2351	19	45,040,933	45,893,307	375	AD	HDL	−0.29	0.09	3.75 × 10^−10^	−0.29
962	6	32,208,902	32,454,577	538	AD	LDL	0.64	0.41	1.69 × 10^−4^	
964	6	32,539,568	32,586,784	26	AD	LDL	0.34	0.11	1.14 × 10^−3^	
966	6	32,629,240	32,682,213	161	AD	LDL	0.76	0.58	1.72 × 10^−5^	
2351	19	45,040,933	45,893,307	369	AD	LDL	0.34	0.11	2.21 × 10^−100^	0.52
2351	19	45,040,933	45,893,307	371	AD	Triglycerides	0.26	0.07	1.02 × 10^−4^	0.26
964	6	32,539,568	32,586,784	26	AD	Total cholesterol	0.41	0.17	1.22 × 10^−3^	
966	6	32,629,240	32,682,213	161	AD	Total cholesterol	0.51	0.26	3.85 × 10^−4^	
1351	8	125,453,323	126,766,827	1102	AD	Total cholesterol	0.30	0.09	1.04 × 10^−79^	
2351	19	45,040,933	45,893,307	373	AD	Total cholesterol	0.38	0.14	3.86 × 10^−9^	0.40
965	6	32,586,785	32,629,239	651	AD	Angina pectoris	0.34	0.12	2.69 × 10^−4^	
2351	19	45,040,933	45,893,307	2620	AD	Angina pectoris	0.37	0.14	1.29 × 10^−10^	0.35
963	6	32,454,578	32,539,567	89	AD	Cardiac dysrhythmias	−0.38	0.14	7.25 × 10^−6^	−0.38
2351	19	45,040,933	45,893,307	2620	AD	Coronary arteriosclerosis	0.53	0.28	9.80 × 10^−28^	0.53
2209	17	45,883,902	47,516,224	4658	AD	Ischemic heart disease	0.33	0.11	1.28 × 10^−3^	
2351	19	45,040,933	45,893,307	2620	AD	Ischemic heart disease	0.44	0.19	6.70 × 10^−17^	0.38
964	6	32,539,568	32,586,784	496	AD	Myocardial infarction	0.41	0.16	8.95 × 10^−4^	
2351	19	45,040,933	45,893,307	2620	AD	Myocardial infarction	0.45	0.20	3.37 × 10^−14^	0.43

AD: Alzheimer’s disease; CAD: coronary artery disease; Chr: chromosome; HDL: high-density lipoprotein; LAVA: local analysis of [co]variant associations; LDL: low-density lipoprotein; *p*: ρ-value; SNP (n): total number of single-nucleotide polymorphisms.

## Data Availability

The GWAS data utilised in this investigation were obtained from publicly available repositories, research groups, or consortia, as detailed in the data sources section. Additional information regarding these datasets, including relevant links to their sources, is provided in Appendix A. Any data generated during the current study are included within the published article and its Appendix A.

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
