# Peer review of "Investigating Genetic Overlap between Alzheimer’s Disease, Lipids, and Coronary Artery Disease: A Large-Scale Genome-Wide Cross Trait Analysis"

_ijms, 2024, doi:10.3390/ijms25168814_

Round 1
Reviewer 1 Report (New Reviewer)
Comments and Suggestions for Authors
From a large-scale genome-wide cross-trait analysis, it is interesting to investigate the genetic overlap between AD, CAD, and lipids. The authors analyzed the database using three methods, indicating a strong correlation between the diseases. They also provide the target genes for the diseases, which will interest the readers. I recommend that the manuscript is suitable for publication in the International Journal of Molecular Sciences.
Author Response
Please see the attachment

Reviewer 2 Report (New Reviewer)
Comments and Suggestions for Authors
The manuscript is interesting to readers in its current form, is a timely study but can be improved:
More details of stats should be provided.
Fig 3 needs more explanation.
Discussion can be more scholarly.
More citations can be included.
Each table of data should include details about study in legend.
Comments on the Quality of English LanguageThe english is fine, may be little improvement is possible.
Author Response
Please see the attachment

This manuscript is a resubmission of an earlier submission. The following is a list of the peer review reports and author responses from that submission.
Round 1
Reviewer 1 Report
Comments and Suggestions for Authors
Alzheimer's disease (AD) is a common neurodegenerative disorder marked by cognitive decline and memory loss. There is evidence linking abnormal lipid metabolism with an increased risk of Alzheimer's disease (AD), and observational studies also suggest a comorbid relationship between AD and coronary artery disease (CAD). In this study, they systematically assessed the genetic overlap between AD, 13 representative lipid traits (from eight lipid classes), and seven CAD traits using large-scale genetic data and robust analytical methods. Main findings include existence of genetic overlap between AD, specific lipids, and CAD traits, implicating shared genetic susceptibility. The identified pleiotropic hotspots are valuable targets for further investigation into the mechanisms underlying AD and its potential comorbidity with CAD traits.
Comments:
The limitations for the study include the results are not very clearly presented the genes are not mentioned as only numbers some clarity on individual genes are needed to improve the manuscript
Reviewer 2 Report
Comments and Suggestions for Authors
My suggestions:
1. In the introduction, I would mention a few CAD risk factor gene examples besides APOE. Also, I would mention briefly the possible proteomic markers of CAD.
2. I would mention SNPs, which may affect the levels of lipidomic markers in the case of AD and CAD.
3. Is there any correlation between lipid markers and the degree of neurodegeneration?
4. Are lipid markers possible markers to predict the risk for dementia and CAD in non-demented individuals in the future?
5. The authors may mention genetic variants and genes, which may be associated with AD and CAD.
Round 2
Reviewer 2 Report
Comments and Suggestions for Authors
The authors fulfilled my suggestions. Thank you.